# Serious Gaming for Behaviour Change: A Systematic Review

**Ramy Hammady** [1,2,*] and **Sylvester Arnab** [3]

1 School of Computer Science and Electronic Engineering, University of Essex, Colchester CO4 3SQ, UK
2 Faculty of Applied Arts, Helwan University, Giza 11795, Egypt
3 Disruptive Media Learning Lab at Coventry University, Coventry CV1 5DD, UK; aa8110@coventry.ac.uk
* Correspondence: ramy.hammady@solent.ac.uk or ramy_hammady@a-arts.helwan.edu.eg or r.hammady@essex.ac.uk; Tel.: +44-7503663334

**Abstract:** Over the years, there has been a significant increase in the adoption of game-based interventions for behaviour change associated with many fields such as health, education, and psychology. This is due to the significance of the players' intrinsic motivation that is naturally generated to play games and the substantial impact they can have on players. Many review papers measure the effectiveness of the use of gaming on changing behaviours; however, these studies neglect the game features involved in the game design process, which have an impact of stimulating behaviour change. Therefore, this paper aimed to identify game design mechanics and features that are reported to commonly influence behaviour change during and/or after the interventions. This paper identified key theories of behaviour change that inform the game design process, providing insights that can be adopted by game designers for informing considerations on the use of game features for moderating behaviour in their own games.

**Keywords:** behaviour change; game design; serious gaming; gamification; review

## 1. Introduction

In the last four decades, games have been designed purposely for behavioural change [1,2]. This is due to their popularity in providing leisure activity and entertainment to players [3]. Researchers exploited the appeal of playing games to influence players' behaviours after witnessing significant behaviour changes while playing [3–5]. Since then, specific games have been particularly created and developed to help transform and change certain behaviours in different fields such as health, psychology, education, marketing, and tourism, etc. [6,7]. New terms have emerged to define these types of games such as: serious gaming, gamification, persuasive games, etc. Designing a persuasive game for serious purposes requires a thorough understanding of the relevant behaviour change theories that can feed into the design process [5]. Games that have been employed to make an impact on players' behaviours are often aided by well-known behavioural change theories and the engaging characteristics of game design elements and mechanics. This manuscript therefore aimed to review and synthesise existing studies on these types of games and their effectiveness/efficacy, the behavioural change theories that are frequently adopted, and the game elements and mechanics utilised to structure these games. This will provide insights into the most frequently applied game elements in these games. This review specifically focused on computer and video games and gamified systems that apply game elements.

The next sections describe terminologies associated to the game types covered by the studies discussed in the paper, which will provide rationale to the systematic review for investigating existing studies associated with games for behaviour change.

### 1.1. Serious Games & Gamification

Serious games (SG) are defined as "interactive computer applications, with or without a significant hardware component, that have challenging goals, are fun to play and engaging, incorporate some scoring mechanism, and supply the user with skills, knowledge or

attitudes useful in reality" *p28* [8]. Another definition relevant to this research is "Games designed for educational, training or modification of behaviour" [9]. SG are also defined as games designed for purposes other than entertainment [3]. They can provide an ideal environment to stimulate players to make decisions in complex processes or change their attitudes or behaviours [10]. SG have applications in a wide spectrum of fields, such as health, the military, education, and government [11]. Researchers in education consider the content of the teaching material used by teachers as the "seriousness" part of these games [12].

Gamification is a term coined in 2008 in the digital media industry [13]. Before the second half of 2010, parallel terms with the same concept have been widely adopted such as "behavioural games" [14], "playful design" [15], "surveillance entertainment" [16], "productivity games" [17], and "funware [18]". Gamification is defined as follows: "Effective gamification is influencing human behaviour through engaging experiences, using game design principles in decision-making applications and services" [19]. Others have defined gamification as the adoption of the methods of game design and game technology outside of the games industry [20]. It is often associated with behavioural economics, where game elements can be used to promote certain behaviour. Its application has spread rapidly in technology and health domains.

Though both terms, gamification and serious games, have been frequently used together in many studies, they are different approaches in terms of the way they work. Gamification blends the game mechanics with traditional activities, e.g., learning, physical activity, etc. [21]. However, serious games follow the typical game structure but also aim to achieve the goal that the game is built for, such as changing attitudes or behaviours. Serious games are entertaining, fun, and interactive even after aligning these games with the mission of achieving the goal needed [22]. While gamification involves the structure of a gamified traditional method, serious games can exist independently [23]. A very comprehensive study concluded that the differences between the two terms are that serious games include all game elements but to varying degrees, while gamification can involve an extraction and the application of a particular game element to a non-game process [24]. This study also criticised considering both terms as complementary approaches, even if they yielded the same game elements, as they applied them differently.

*1.2. Definition of Behaviour Change Games*

Behaviour Change Games (BCG) form a subset of serious games, which were designed in order to support attitude and change behaviours, as stated by Boyle and Connolly [3]. BCG have the same nature as what are often called persuasive games [25]. Persuasive technology is defined as "an interactive product designed to change attitudes or behaviours by making desired outcomes easier to achieve" [26]. Naturally, video games are used to implement persuasive strategies by utilising the power of mechanics and elements of the game design; for instance, self-monitoring, which can allow people to monitor themselves; conditioning, which offers rewards based on the performance of particular behaviour; and tunnelling, which is about leading players through a prearranged sequence of actions to either encourage or discourage particular behaviour [27]. Persuasive games have applications in many fields, such as health games, political and social games, and advertising games aiming to change behaviour regarding certain issues, such as encouraging recycling, discouraging smoking, or increasing voting [25]. Considering the significance of these types of games, this review focused on BCG that have been produced in different areas.

*1.3. Game Design Elements and Behaviour Change*

Werbach and Hunter [28] categorised game features into Mechanics and Components. Bharathi and Singh [29] listed game design features, which are relevant to Mechanics, e.g., challenges, feedback, and rewards; and Components, e.g., achievements, avatars, badges, leaderboards, levels, points, and social graphs, as well as latent game design features.

Alike entertainment games, serious games have generic components and game elements, which are replicated and used in different titles [30]. However, the usage of these elements is for scaffolding purposes, and not as they are normally used in entertainment games [31]. Generally, features and mechanics in SGs have not yet been characterised and defined due to their complex nature compared with entertainment games [30]. To be more focused on behaviour change games or persuasive games, game design elements and mechanics vary based on the type of gamified application and the desired outcome, i.e., whether they are built for learning purposes, health behaviours, or fitness purposes.

For instance, in health applications, the game design incorporates several strategies, such as monitoring, harmony, group opinion, and dis-establishing [32]. In fitness applications, game design has been used to encourage players' long-term engagement through social interaction. Therefore, different elements and mechanics such as social play, micro goals, fair play, and marginal challenge are often employed [33].

*1.4. Behaviour Change Theories and Effective Game Design*

Various persuasive games or BCGs from different fields are informed by the theories of behaviour change in the process of game design, such as health games [5] and exergames [34]. There is a relationship between the theories of behaviour change and the choice of game elements that should be embedded in BCGs.

Behaviour change through games often relies on Social Cognitive Theory (SCT) and the Elaboration Likelihood Model (ELM), comprising four main steps that demonstrate information processing, i.e., attention, retention, production, and motivation [5]. The first two phases are relevant to the learning process, and the second two are relevant to performance [35].

The Theory of Planned Behaviour (TPB) is one of the most frequently cited theories for predicting human social behaviour [36,37]. It was adopted to design games to influence the public's behavioural intention to play online games [38]. Social-participation theory relies on involving a large group of people in an activity, and it has been used to inspire persuasive technology and game design processes [39].

Flow theory [40] has been used to improve interactive experience and video game design [41]. When the difficulty level of a task is balanced with competency, individuals can be more focused on achieving goals and more immersed in an activity, with a feeling of pure pleasure and enjoyment, without the need for external rewards [42]. Flow theory has also been used to drive behavioural change in games, and it is considered a natural foundation of games, especially educational games [43].

*1.5. Problem Statement*

Existing studies and reviews often report the effectiveness of game interventions on behaviour change without offering any insights into why and how games and gameplay are effective on a granular design level by reflecting on the choice of game elements used in the design. They often did not consider the strategies for selecting game elements or adopting behaviour change theories, or both combined, to structure the game design process [44–49]. Despite the rapid growth of persuasive games in many fields, there is a lack of studies that articulate the game design process and the most frequently used game design elements in BCGs. Moreover, they are fragmented across different studies [8,50], which focus on designing serious games generally. This incoherent representation of the knowledge around game elements and their role in the design of behaviour change games has motivated us to extract, analyse, and synthesise the game design elements that are frequently applied to and/or mentioned in several studies in all fields that have adopted BCGs and persuasive games. This paper aimed to provide insights into game elements that were frequently applied in the games that have been reported to have a significant effect on players' behaviour. The findings will provide BCG and persuasive games designers with additional insights on game design elements and mechanics, associated with the behaviour change theories they could consider to enhance the impact of the games they are designing.

## 2. Methods

### 2.1. Identifying Terms and Databases

Search keywords were input through some databases, such as Science Direct, ACM Digital Library, and Taylor and Francis Online. These databases are comprehensive and diverse, covering many fields. They also contain publications from 1950 until 2018. However, this research only focused on the last 25 years, around the time serious games have been in application. The following keywords were used: ("gamification") OR ("Serious game") OR ("Persuasive games") OR ("Behaviour change games") OR ("video game*") OR ("gamified") OR ("game element") OR ("game design") OR ("Computer games") AND ("change attitude" or persuasion" or "behaviour"). This search resulted in 3352 articles.

### 2.2. Selection of Papers Method for the Review

After considering the keywords in the search procedure through the targeted databases, only 205 articles were considered based on the inclusion criteria for the abstract. The inclusion criteria included the following: (a) included a game as evidence to change behaviour; (b) included game design demonstration features, components, elements, mechanics, or latent features; (c) concluded a positive, rather than negative, influence of persuasive games or serious games; and (d) studies within the period from 1 January 1993, to 31 December 2020. The review process excluded papers not including any empirical studies or games developed for the sake of the research, as shown in Figure 1.

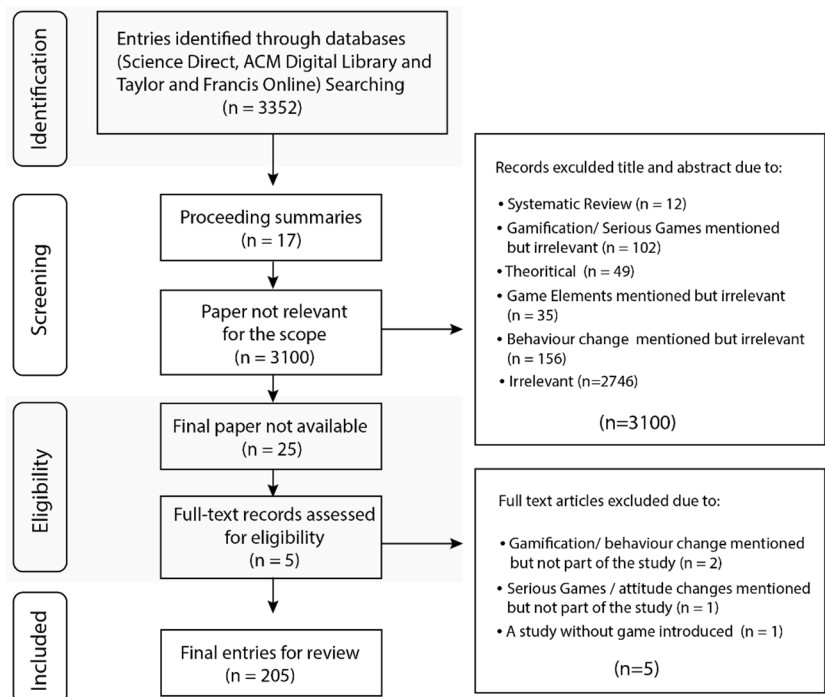

**Figure 1.** Flowchart of the literature review process.

## 3. Data Analysis

The 205 papers that met the criteria were analysed to investigate the most common game elements—either game mechanics, game components, or latent features—as well as the theories of behaviour change adopted for designing the games of the explored studies. In line with other studies [29,51,52], we followed the Werbach and Hunter [28] classification of game elements by mechanics and components.

### 3.1. Categorisation of Games

This study categorised games that were designed to change behaviours and attitudes as a separate category away from the mainstream of normal entertainment games. Thus,

this systematic review explored studies that adopted BCG in many fields such as health, education, energy, urban, and engineering. The scope also extended to include serious games and gamified systems or applications which implemented game elements, components, or mechanics that have had an impact or influence on behaviour change in one way or another. Moreover, the review included studies that adopted the theories of behaviour change to design and build the game of the study.

### 3.2. Categorisation of Objects

The main objective of this literature review was to determine whether published games aimed to change behaviour, and through which design choices. However, the categorisation included other relevant objectives, such as enhancing motivation, changing (encouraging/discouraging) habits, changing attitudes about particular subjects, or enhancing people's perceptions, which are the subsets or precursors to behaviour change. Additionally, this review only focused on the positive influences of games and did not consider studies on the negative influences on human behaviours.

Our study considered games for any platform—desktop computers, videogame consoles, mobile phones, tablets, online games, arcade games, and head-mounted display games.

## 4. Results

### Generic Findings

Behaviour change games have increased in popularity during the last decade. Figure 2 demonstrates the publication trend in studies focussing on BCG from 2010, which peaked in 2013.

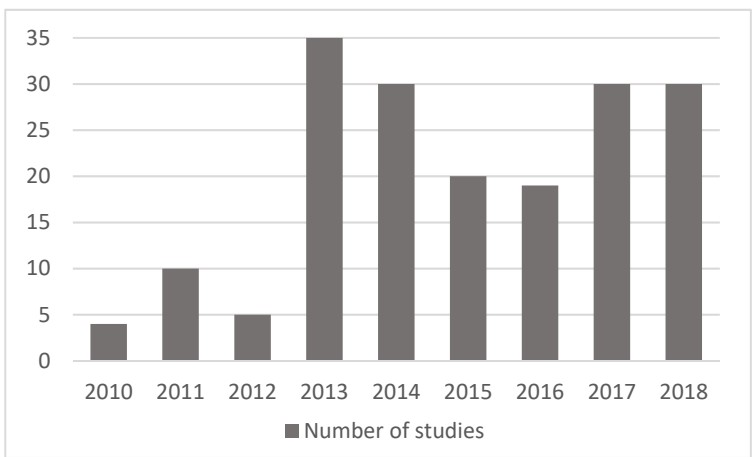

**Figure 2.** The number of Behaviour Change Games (BCG) studies by year.

BCG have emerged in many fields; as seen in Figure 3, results from the 205 studies showed high frequencies of BCG in three fields: Psychology (54), Health (40), and Education (37). Other fields include Energy (9), Business (8), and Information Systems (6). A few studies were also found in the Engineering, Transportation, Climate Change, Management, Safety, Tourism, Marketing, and Automotive fields.

Our review quantified the number of times that specific game elements were explicitly mentioned. As depicted in Table 1, the game features were classified according to [29], and the table also shows some of the studies that involved them. For more details, please read Table S1 in the supplementary attachment to the article that demonstrates the classification and definitions of game elements. There are game features that show the high frequencies among all game features such as rewards, challenges, points/scoring, and feedback (as shown in Figure 4), followed by other features, such as competition, levels, avatars, and eliciting action/decision. Analysis of this mapping allowed further investigation into the

rationale behind the adoption of these elements in the fields that have shown the highest adoption of BCG studies.

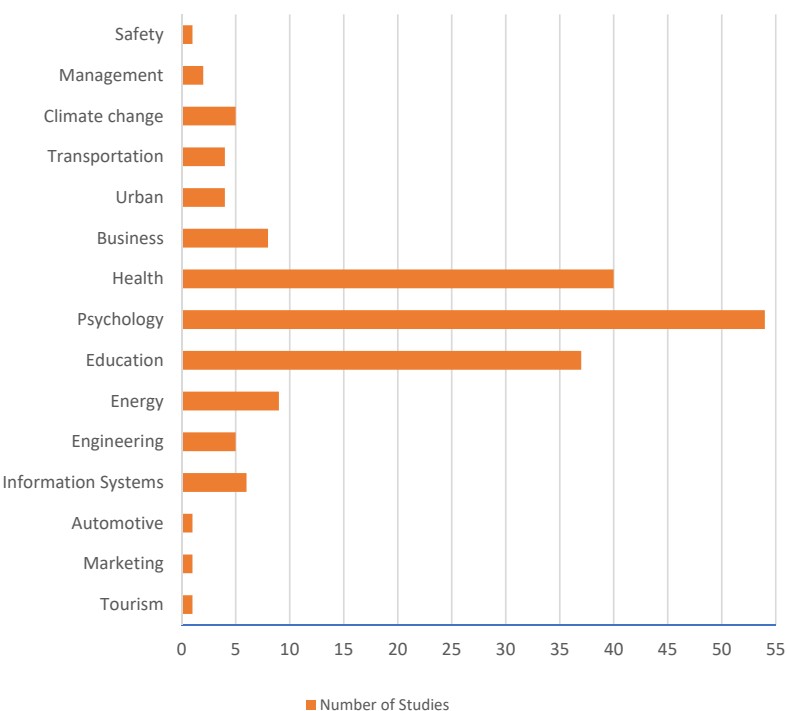

**Figure 3.** The BCG paper frequencies in application domains.

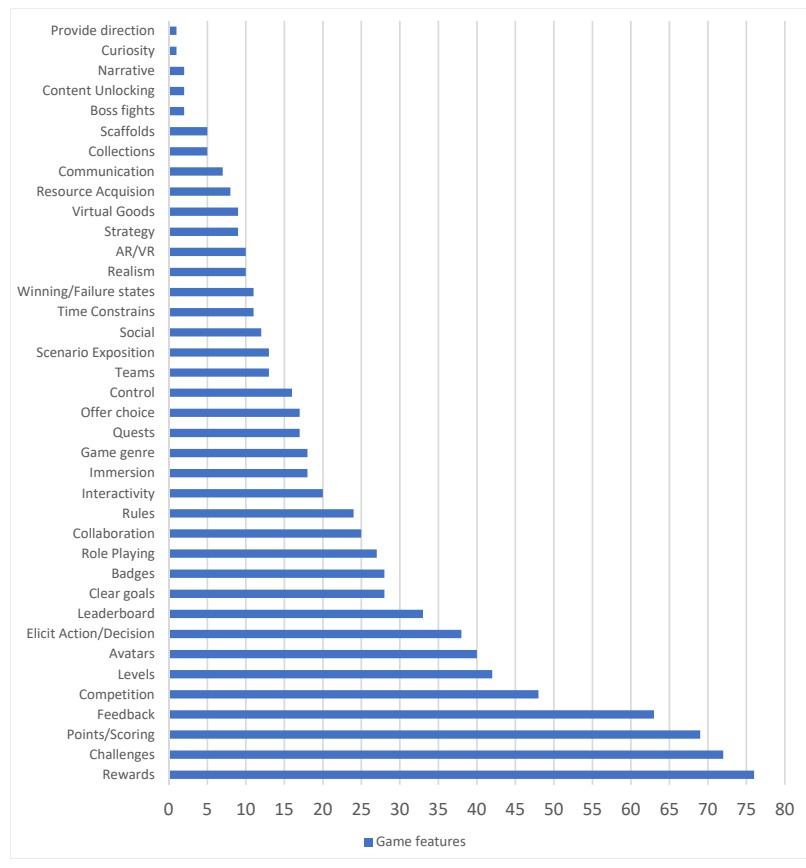

**Figure 4.** Frequencies of game elements and features from the analysed literature.

**Table 1.** Classification of game elements and their frequencies in BCG studies.

| Category | Game Elements | Frequencies | Studies (e.g.,) |
|---|---|---|---|
| Mechanics | Challenges | 72 | [53–56] |
| | Feedback | 63 | [53,57–59] |
| | Rewards | 76 | [60–62] |
| | Competition | 48 | [63–66] |
| | Resource acquisition | 8 | [53,67–69] |
| | Wining/failure states | 11 | [57,61,70,71] |
| Components | Achievements | 20 | [60,72–74] |
| | Avatars | 40 | [53,61,75] |
| | Badges | 28 | [76–79] |
| | Leaderboard | 33 | [80–83] |
| | Levels | 42 | [84–86] |
| | Points | 69 | [53,87–89] |
| Latent Features | Boss Fights | 2 | [90,91] |
| | Collections | 5 | [90,92] |
| | Content unlocking | 2 | [68,93] |
| | Gifting | 4 | [94,95] |
| | Quests | 17 | [53,61,96] |
| | Teams | 13 | [97,98] |
| | Virtual Goods | 9 | [99–101] |
| Other Game Features | Rules | 24 | [102,103] |
| | Scaffolds | 5 | [104,105] |
| | Curiosity | 1 | [106] |
| | Immersion | 18 | [91,104,107] |
| | Social | 12 | [62,108,109] |
| | Scenario exposition | 13 | [110–112] |
| | Problem Setup | 1 | [112] |
| | Offer Choice | 17 | [112–114] |
| | Provide Direction | 1 | [115] |
| | Elicit Action/Decision | 38 | [88,116] |
| | Time constraint | 11 | [103,110,117] |
| | Game genre | 18 | [118,119] |
| | Role Playing | 27 | [120,121] |
| | Discovery | 1 | [122] |
| | Augmented/Virtual reality | 10 | [59,123,124] |
| | Control | 16 | [78,85,125] |
| | Interactivity | 20 | [105,126] |
| | Realism | 10 | [91,122,127] |
| | Narrative | 2 | [128,129] |
| | Strategy | 9 | [6,130,131] |
| | Collaboration | 25 | [70,132,133] |
| | Clear goals | 28 | [53,63,134,135] |
| | Communication | 7 | [62,102] |

The systematic review also investigated the theories adopted during designing the behaviour change games, and it revealed a number of behaviour change theories and other game design theories, such as the theory of flow. These theories allow for the investigation into how players' behaviours were stimulated in the study [102,136]. 'Self-Determination Theory' was the most cited and adopted theory in BCG studies, as shown in Table 2. This was followed by the 'Theory of Planned Behaviour', 'Social Cognitive Theory', and 'Motivational Theory'.

**Table 2.** Adopted theories and their frequencies in BCG studies.

| Theories | Frequency | Examples of Studies Which Adopt the Theory |
|---|---|---|
| Self-Determination Theory | 34 | [57,58,102] |
| Theory of Planned Behaviour | 13 | [73,78,121] |
| Social Cognitive Theory | 11 | [59,136,137] |
| Motivational Theory | 9 | [78,103,108] |
| Flow Theory | 8 | [59,115,138] |
| Theory of Fun | 5 | [112,133,139] |
| Cognitive Evaluation Theory | 3 | [103,140,141] |
| Self-Perception Theory | 3 | [91,113,117] |
| Elaboration Likelihood Model | 1 | [136] |

## 5. General Findings

### 5.1. Literature of Game Features

Some game features are mentioned and adopted repeatedly in many studies, implying their significance in behaviour change games.

We begin with 'rewards', the most frequently discovered game feature in the review. This feature is defined as 'some benefits that go together for some action or achievement in the Game' [29]. To justify the prominence of a reward system among other game features, Skinner [142] believed that extrinsic rewards can significantly control behaviour. He found that adopting a motivation system with certain behaviour makes it more likely that this behaviour would be emitted and the effect stays as long as the motivation system is applied. Then, by terminating the reward system, this particular behaviour comes back to the baseline. The reward system is thus a motivational strategy that has a significant influence on behaviour [143]. Deci [144] found that it has a significant effect on self-determination (i.e., autonomy) and competence. That was based on the interpretations of rewarding the recipients and what they feel. However, types of rewards can be either intrinsic or extrinsic, and the effect of whether someone will self-regulate based on their extrinsic motivation differs. Meaningful rewards can influence intrinsic motivation towards a particular behaviour.

The second repeated game feature, 'Challenge', was simply defined by Bharathi and Singh [29] as 'Puzzles or other tasks that require effort to solve' or 'to increase the level of difficulty' [5]. According to Baranowski [69], the challenge technique has been used to change a particular players' behaviour. It also works concomitantly with a reward system; for example, when players solve a complex challenge, they will be rewarded. Challenging a player follows the strategy of deferral, which increases the amount of time spent playing the game [145]. Additionally, using a challenge feature in games gives players a sense of accomplishment [146] and leads to increased competence, which accordingly arouses feelings of fun and enjoyment [69]. Challenge is also one of the theories of flow factors [147], namely, to ensure the consistency of keeping the player in the flow status is to balance the levels of challenge and competence. Therefore, as long as the player is kept in the flow of playing, the likelihood of behaviour changing is higher.

The third most frequent game feature is 'Points/Scoring', defined by [29] as *'Numerical representation of game progression'*. A scoring system as a game component is considered as one of the strategies that engage players more in the game [148]. It makes more sense when other players are involved, as it creates the sense and value of competence [149]. So, players can compare scores, giving them a sense of accomplishment, similar to a reward system [150]. It can also be used as a strategy to increase game play motivation in pursuit of short-term rewards. [151] considers it as a short-term reward. It even encourages players to repeat the task to achieve a higher score. Therefore, adding a scoring system to games or gamified systems is useful for engaging players to play more and motivating them to perform certain desired behaviours or tasks.

The fourth most frequent game feature was 'Feedback', defined as *'Information about how the player is doing'* [29]. Immediate feedback has also been associated with clear goals and challenges feature to keep the player in the flow experience of the game [147,152]. Feedback in games or gamified systems has an influence on behavioural intention [152]. Moreover, with the presence of feedback, it is possible to trigger some activities [153]. It can enhance players' performance in a particular task, which fares better when compared with games without feedback features [154].

### 5.2. Theories of Behaviour Change

Among all fields, the most adopted theory for designing BCG was 'Self-Determination Theory' (SDT), as it is empirically based on human motivation [155]. SDT as a human motivational study looks for ways to energize and direct human behaviour [156]. In fact, SDT concerns personal development, psychology needs, self-regulation, energy, and the influence of the social environment on motivation [155]. It also addresses autonomy by

investigating the sense of stimulating or enthusiasm for doing tasks [157,158]. SDT emerged three decades ago and became widespread, entering several practical fields, such as sports, education, and health, due to its influence on human behaviour [155]. SDT is involved in many games studies, as it was used to investigate players' motivations and choices during play [159]. Other studies investigated game features associated with the SDT basic needs such as autonomy and competence to influence the game's enjoyment and motivation for future play [160]. SDT in games can explain why individuals choose to participate and apply extra effort, as it is considered an intrinsic motivation.

The second most adopted theory was the 'Theory of Planned Behaviour' (TPB), which was coined by Ajzen [161] and became one of the most cited and influential theories for predicting human social behaviour [37]. There are three beliefs in this theory that lead to desired behaviour: behavioural beliefs, normative beliefs, and control beliefs [161]. TPB is an influential theory of behaviour change and attitude as it is a significant predictor of people's intention to behave in a specific way [3]. TPB has frequently been adopted in game studies for understanding their intentions to perform certain actions in games or play games in general. For instance, it was used for understanding the behavioural intention to play online games [38,162,163]. Another study used TPB to explain why game players tend to buy virtual goods in virtual worlds with real money [164]. TPB was also utilised by a study investigating the reasons why adolescents perform risky driving behaviours while playing racing video games [165]. Thus, BCG adopts TPB not only to investigate certain behaviours and understand them, but also to predict and control certain behaviours.

Social Cognitive Theory (SCT) was the third most frequently adopted behaviour change theory in BCG studies and was the most commonly cited theory that introduces a foundation for behaviour change [166]. SCT defines behaviour change as a function of improved skills and confidence or self-efficiency in performing the new behaviour [5]. It identifies a set of determinants, the way in which they work, and efficient ways to translate this knowledge into effective practices [167]. Adopting SCT in game studies can help explain why players perform certain actions or identify the purpose of playing particular games [168,169]. SCT also aims to shape people's behaviour during gameplay [170].

## 6. Findings on Game Features in Relevant Fields

The outputs of the general exploration of the most adopted game elements and behaviour change theories in BCG studies have not been scoped to a specific domain, as the studies appear across the various fields as shown in Figure 3. The general findings as discussed cannot, therefore, represent a particular domain due to the diversity of the contexts and the motivation from one domain to another.

This section provides an overview of the studies within the domains that frequently adopt BCG, according to the most frequently applied game features and theories. The three key domains identified (Figure 3) were health, education, and psychology, which were also reflected by the focus of existing reviews of the outcomes and effectiveness of serious gaming in several fields such as in the health domain [171] and in learning and skill enhancement and engagement [172]. The next section demonstrates the implications of using BCG particularly in the mentioned fields. The next sections outline the motivations behind using BCG and provide insights into the considerations relevant for practitioners and academics in the in the fields of health, psychology, and education.

### 6.1. Literature in the Health Domain

Serious gaming is widely used as an intervention in medical and health domains in order to change players' behaviour. Indeed, numerous games using the term 'Health Education' have been created as interventions to influence players towards healthy performance. Early research showed positive results when they utilised health educational computer games for enhancing the performance of adolescents [173,174].

Games have been used to help regulate behaviour to achieve healthy outcomes. For instance, *'Packy & Marlon'* is a game that was utilised to improve the self-care of children

and adolescents with diabetes [175]. A similar purpose was reported by another study using a board game called *'Kallèdo'* [176]. Another study developed and evaluated *'Right-Way Café'*, a game intended to promote a nutritious and healthy diet, which has shown high commitments from students in the treatment group regarding healthy eating habits compared with those in a control group [83]. A similar study was conducted by developing the game '*Squire's Quest*' for children as an intervention to increase their consumption of fruits, juices, and vegetables [177]. Another serious game was utilised as an intervention to maintain children's fruit and vegetable intake [178]. Some games, known as exergames, have also been used to motivate players to participate in physical activity [179].

Games have also demonstrated their impact on enhancing the behaviours of people who have alcohol obsession [180]. Some scholars explored using computer games as interventions as part of cognitive behavioural therapy [181]. A contribution in enhancing mental health and wellbeing for young men has also been demonstrated through engagement with serious gaming [182].

Figure 5 explores the most frequently adopted game features in the health domain. 'Challenges' was the most common game feature, followed by 'Rewards', 'Points/Scoring', 'Feedback', and 'Levels'.

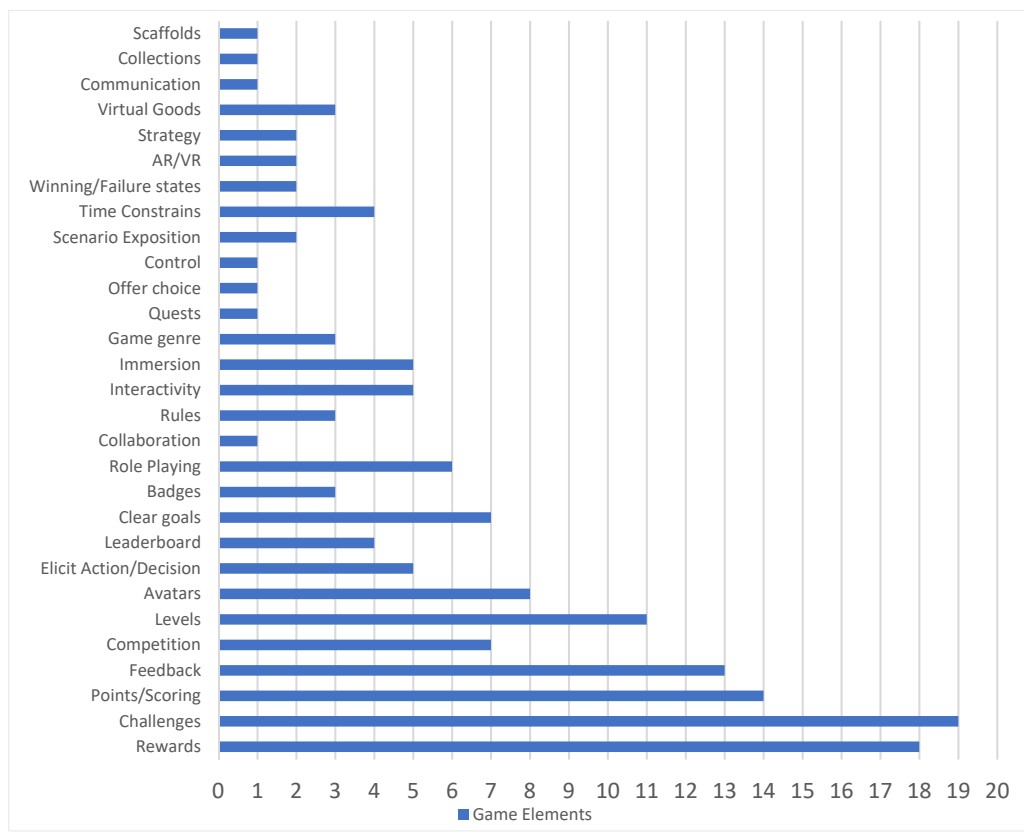

**Figure 5.** The frequencies of game features in the Health domain.

With regards to behaviour change theories emerging from the studies associated with the health domain, we found that social cognitive theory was mostly quoted, followed by self-determination theory, the theory of planned behaviour, and flow theory, as depicted in Figure 6.

### 6.2. Literature in the Psychology Domain

The psychology domain underpins studies that focus on attitudes and behaviours, mental wellbeing, brain cognition, social behaviours, environmental behaviours, personality, and motivation. Serious games used as interventions to change human behaviour can be part of a psychotherapeutic process [183]. Games, when applied as a persuasive

technology, commonly use incentives to engage individuals in healthy behaviour [184]. Persuasive games used to change players' behaviour include the 'Smoke' game [27] and quitting smoking games [32]. Serious games were also utilised to promote physical education, also known as 'exergames' [185]. Games have also been reported to help users learn when and what to avoid when it comes to phishing attacks, using a stimulus to nurture informed behaviour [186]. Some studies state that playing games with prosocial content can enhance prosocial behaviour [187–189], where games could achieve a transference of intimacy motives and satisfaction between the players in the virtual world and their real world spouses [190]. Computer games have also been proven to enhance social skills; Kowert and Oldmeadow [191] articulated that the amount of involvement in online games has a positive relationship with emotional expressivity and emotional control.

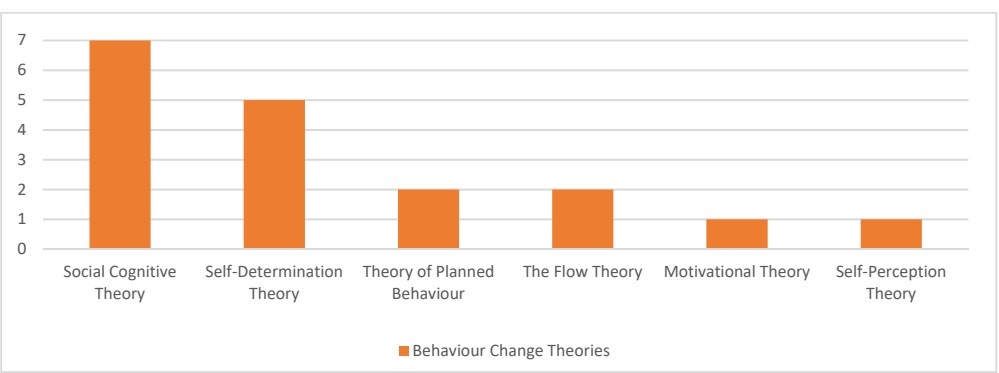

**Figure 6.** The frequencies of Behaviour Change theories in the Health domain.

By surveying the game features based on the studies associated with the psychology domain, it was found that Rewards was the most popular, followed by Challenges, Points/Scoring, and Competitions and Avatars, as summarised in Figure 7.

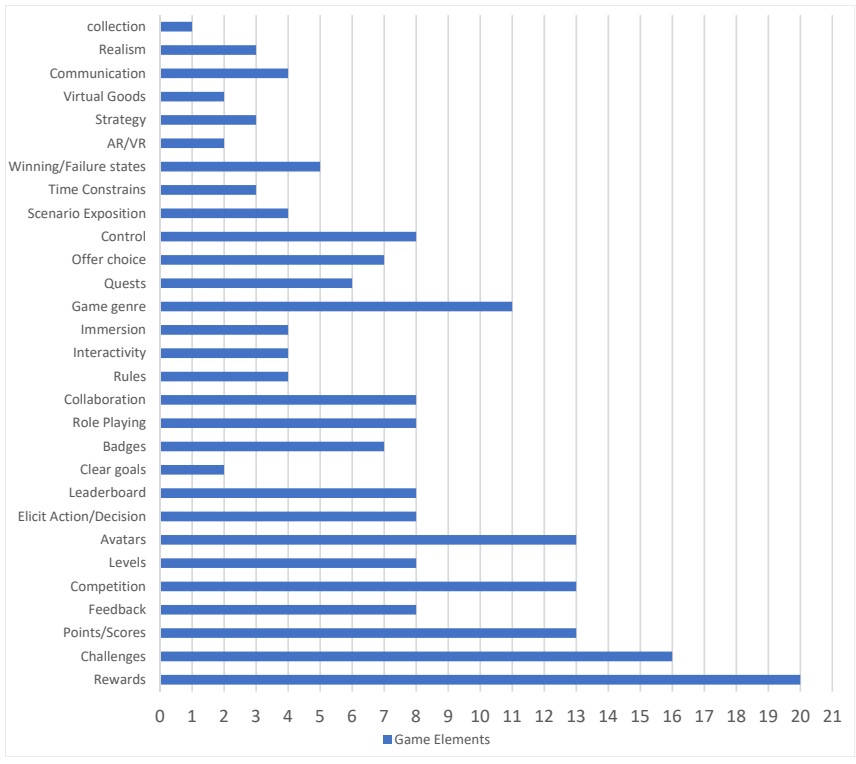

**Figure 7.** Frequency of game features in the Psychology domain.

In the investigation into the frequency of adoption of the behaviour change theory, we found that self-determination theory was the most adopted, followed by the theory of planned behaviour, social cognitive theory, and flow theory, as depicted in Figure 8.

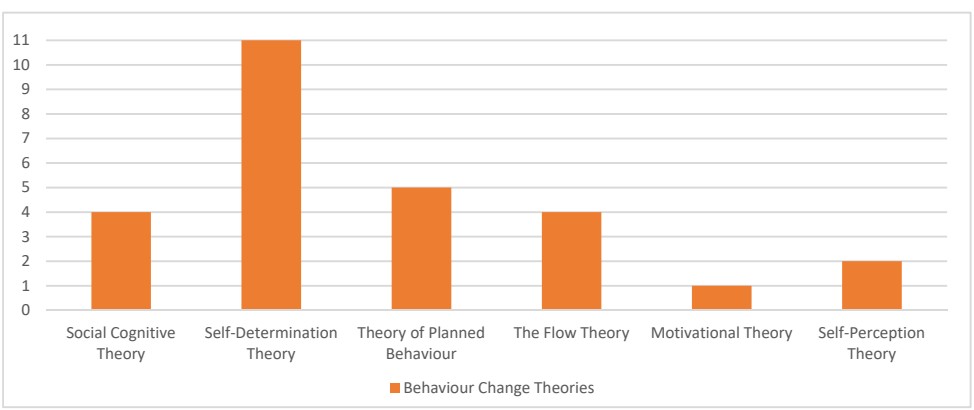

**Figure 8.** Frequency of behaviour change theories in the Psychology domain.

### 6.3. Literature in the Education Domain

The applied use of games is increasing within the education community, building on the rationale that young people are very familiar with games and gameplay, which opens up opportunities for using games to present and deliver educational content [192].

In recent years, practices in academic teaching have witnessed the use of multimedia technologies such as video games and serious games to enhance practical skills in schools [193]. A recent significant critical review study showed the effectiveness of using the educational games compared with the traditional methods in addition to promoting attitude change [194]. Moreover, many studies suggest that serious gaming can stimulate cognitive aptitude [195].

Several studies have examined the effectiveness of serious gaming with regards to knowledge acquisition through several fields, and the results show better performance from players compared with traditional instruments. Arnab [122] found that serious games, when used within a classroom setting as a teacher-facilitated instrument, were able to encourage communal discourse and reflection that led to the nurturing of appropriate attitudes towards understanding the impact of sexual coercion in relationships. Online computer games have also been shown to boost the level of interaction and communication between learners and encourage students to recognise their own problems and find constructive solutions [196]. The interactivity of computer games applied in museums encourages players to invest a considerable mental effort to acquire vocabulary from other languages [197]. Players tend to be more encouraged to make decisions and think analytically and critically in some computer games that support urban planning education [198]. Some educational computer games are able to heighten students' motivation to learn history, such as *Civilization III* [199]. Video games also have a significant impact on the physical education domain, as they stimulate players to participate in physical activity [200]. Numerous studies assess the effect of using computer games practically in various educational fields and find that students are highly motivated and show effectiveness towards curricular objectives [201–203].

Some scholars have attempted to set serious gaming fundamentals for educational purposes by translating learning goals or practices into mechanical elements of gameplay [30], which are then extended to map against the motivational needs based on SDT [204]. Arnab et al.'s (2015) pedagogical-game mapping has also been used to inform the aforementioned game for raising understanding around sexual coercion.

Games also contribute to the environmental education domain; video games have been used to encourage positive attitudes towards environmental sustainability through free choices for school students [205]. Moreover, games have also been developed to transform

players' behaviour towards energy efficiency, such as *'Green My Place'* [206] and *'The Energy Battle'* [207].

The most frequently adopted game features in the education domain, as summarised in Figure 9, were 'Rewards', followed by 'Feedback', 'Challenges, 'Competition', and 'Points/Scores'.

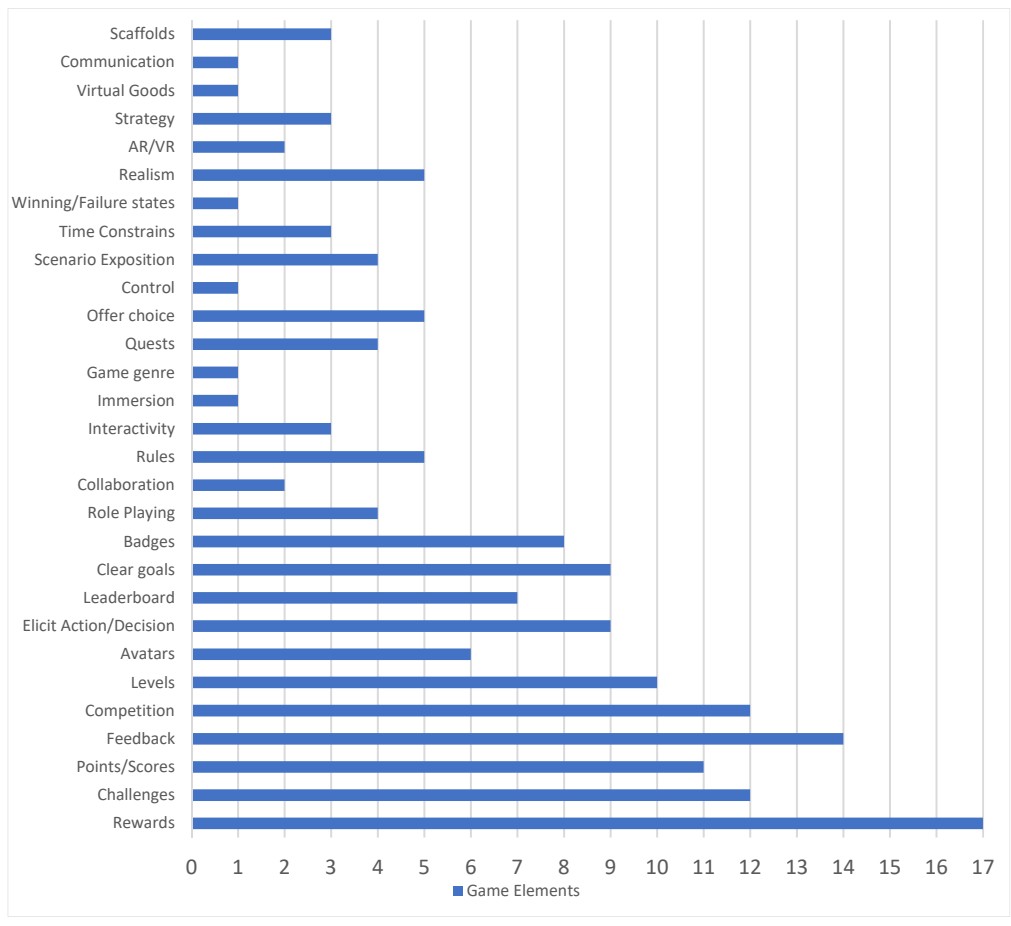

**Figure 9.** The frequencies of game features in the Education domain.

With regards to behaviour change theories, SDT was the most adopted theory by far, followed by theory of planned behaviours, SCT, theory of fun, motivational theory, and cognitive evaluation theory, as depicted in Figure 10.

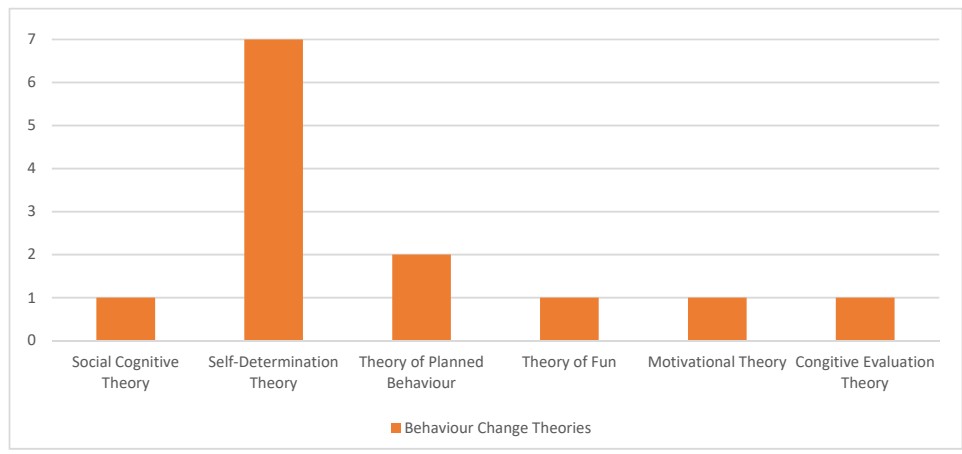

**Figure 10.** Frequency of behaviour change theories in the Education domain.

## 7. Discussion

We presented the general findings based on the frequency of game elements and behavioural change theories adopted in behaviour change studies, followed by an overview of these within the three main domains—Health, Psychology, and Education. The paper highlighted the most applied game elements across the most three main fields to provide insights into the mapping that might occur between these fields and game features and between the fields and behaviour change theories. Table 3 reflects what is presented in Figure 4, i.e., the top game features often found in the studies associated with the 16 application domains matched what was found specifically in the studies in the top three fields. Similarly, the behaviour change theories often adopted in all fields in Table 2 mirror the top influencing theories in Table 3. The contextual differences between the fields and the relationships with the game features and theories are articulated in the following sections.

**Table 3.** Frequency of game features and behaviour change theories among the domains.

| Game Features | Health | Psychology | Education |
| --- | --- | --- | --- |
| Challenges | 19 | 16 | 12 |
| Rewards | 18 | 20 | 17 |
| Points/Scoring | 14 | 13 | 11 |
| Competition | 7 | 13 | 12 |
| Feedback | 13 | 8 | 14 |
| Levels | 11 | 8 | 10 |
| Avatars | 8 | 13 | 6 |
| **Behaviour Change Theories** | **Health** | **Psychology** | **Education** |
| Social Cognitive Theory | 7 | 4 | 1 |
| Self-Determination Theory | 5 | 11 | 7 |
| Theory of Planned Behaviour | 2 | 5 | 2 |
| Theory of Fun | 2 | 4 | 1 |
| Motivational Theory | 1 | 1 | 1 |
| Self-perception Theory | 1 | 2 | 0 |
| Cognitive Evaluation Theory | 0 | 0 | 1 |

### 7.1. Game Features

7.1.1. Health

'*Challenges*' was the most common game feature in health studies; this feature taps into players' tendency to accept challenges in order to onboard and engage, as this behaviour can motivate players to complete the game [54]. Various health studies employ games that apply challenge mechanics that influence players to do or stop certain habits when they play together [87]. Challenges are also essential in exergames to encourage players to improve their skills and not lose the momentum and flow of the activity [84].

The existence of '*Rewards*' is a natural result of the challenge element in health games. Therefore, the element of rewards was the second most common in health games, which is due to it being usually applied in a gamification strategy to encourage individuals to complete the required tasks [208]. Rewards also motivate players to progress and to achieve the objectives of medical games in multiple stages [209]. Ref. [210] found that using rewards produces a sense of achievement for players.

Outcomes from challenges are often visualised as feedback through '*Points/Scoring*', which was the third most common feature in health studies. The adoption of points and scoring in medical applications allows for the identification of problematic areas based on the challenges and scenarios the children engage with within the game, which, subsequently, allows doctors to identify issues faced by children after cancer treatment [211]. Studies using this feature have also shown a positive impact on stopping unhealthy attitudes as an outcome of engaging with mobile games [212], as they can be accompanied by challenges mechanics in order to be used for different measures, e.g., they can be part of a feedback loop to inform players of where they are in the game.

Rani and Sarkar [213] proved through their framework that real-time feedback can maintain the optimum challenge in games. Therefore, *'Feedback'* is a significant feature, which was ranked fourth most common in health games/gamified systems. Feedback has been utilised in health games in the process of behaviour assessment for the sake of enhancing engagement and comprehension, leading to changes in players' behaviour [214].

*'Levels'* represents the continuity of the challenge concept in a game, with every new game level marking a new challenge [215]. Levels were noted to be the fifth most used feature in health studies. Levels have contributed to health game design to scaffold the development of mental health literacy in young people, where each player's skill level is shared with other players to boost the game's appeal by fostering a sense of competition [216].

The sharing of players' statistics often encourages *'Competition'*, which has been considered a key element that promotes an entertaining experience for players [217]. Being the seventh most used in health games, competition is not entirely important for encouraging engagement in health games. However, this element plays a role in maintaining players' interactions during the process towards nurturing and changing behaviours and attitudes. Competition in health games often takes the form of team competitions, which also promotes collaboration, where players cooperate to ensure their team is winning [81].

It is interesting to note that a sense of control through customisability is also important, such as the use of characters controlled by players, which can influence engagement in the game-playing experience [218]. So, considering engagement is a significant factor in changing behaviours [219], the 'Avatars' element was the sixth most popular in health studies. Avatars used in gamified health applications have played a significant role in helping children choose healthier food [220].

Generally, designing games/gamified systems for health studies requires a clear understanding of the aims and objectives of the impact needed. The BCG design for healthy purposes should be composed collectively from several game elements that have shown significant results in changing behaviours. The results of the systematic review, particularly for the health studies samples, showed the priority of involving the challenge element and several ways of rewarding either scores/points, or incentive feedback. Additionally, to engage players within the game for a greater impact, involving competitions between players and/or teams would be effective. Game levels, also part of the challenge concept, and the ability to control avatars, showed a greater influence on player engagement.

### 7.1.2. Psychology

In psychology studies that involve BCG, *'Rewards'* took the first place. Wang and Sun [221] articulated those rewards can deliver pleasure and satisfaction to players. Rewards also showed a positive influence on adding the enjoyable factor for the players [222], which consequently impacts the intrinsic motivation [223]. A psychological study found that rewards in gamification systems can increase customers' loyalty to their vendors [224].

*'Challenges'* was the second highest adopted game feature in this domain, shown to increase the progress of a player in a game [55]. It was also demonstrated to increase players' effectiveness in mastering a game, which accordingly reflects on the players' behaviour [82]. Studies show that challenges are usually implemented with other game features to identify the nature of the players' behaviour [90]. Other psychology studies embed challenges in their games to investigate players' responses to different choices presented in the game [65].

*'Points/Scoring'* took third place in phycology studies. Scoring as a game mechanic was also shown to have a positive impact on attaining the athlete engagement goals [225], which is perhaps caused by the competitive nature and scoring provided the familiar feedback for the players. *'Competition'* took third place, as it demonstrated significant contribution to the understanding of a player's behaviour and to stimulate players for doing more efforts towards certain behaviours during play [226]. *'Avatars'* took third place as they were adopted for exploring the identity of players [227].

*'Feedback'* took fifth place. Psychological games prefer the use of feedback as it was found to motivate players to continue playing or to mitigate their poor performance [141].

Generally, the reviewed psychology BSG studies often prioritise rewards and incentives, as well as the use of challenges. Points/score and feedback are part of the challenge, feedback, and reward loop. This can be combined with the adoption of avatars and the sense of competition in the game mechanics in order to obtain effective outputs.

### 7.1.3. Education

*'Rewards'* took first place and was often demonstrated to influence intrinsic motivations when it comes to learning [228], such as increased or sustained engagement in educational games [229,230].

Education significantly relies on *'Feedback'* [231]; therefore, it took second place in BCG in education studies. Studies have shown that feedback has significant influences on learning effectiveness if learners regularly receive it [44], which perhaps links to the use of points/scoring and rewards as part of the informative feedback.

*'Challenges'*, in third place, was revealed to be slightly less significant in education studies when compared with health and psychology, though this element is still one of the main features in BCG games or gamified education systems for achieving better learning and retention [232]. Challenges also help players stay in the flow towards mastering the game [233]. As mentioned earlier, challenges can stimulate motivation. For instance, Denis and Jouvelot [234] suggested that motivation during an educational process can strengthen students' attention. Furthermore, when challenges are presented in a game, the game provides a safe space for training and can meaningfully evaluate the student's learning performance [235]. The previous elements are often part and parcel of the challenge-feedback-reward loop.

*'Competition'* in an education context can motivate students to learn and change their attitudes drastically [236]. Therefore, this game element took fourth place in education, as it was shown to increase students' abilities to learn programming skills [237]. *'Points/Scoring'* took fifth place in education studies. Games that use this feature amongst others were also shown to motivate learners to focus on their performance to achieve goals, as they seek higher scores during play [154]. *'Levels'* was the fifth most used and mentioned in educational BSG studies. Game levels have been utilised for educational purposes as they support the levelling up approach where learners progress through the game from one level of difficulty to the next, which can enhance their engagement in learning [238]. *'Avatars'* were in ninth place in education studies. The adoption of avatars in BCG games was shown to help players achieve learning goals in education games [239].

Designing BCG for an education context emphasises the use of rewards (points/scores) due to its significance and perhaps familiarity in the common education process, followed by the integration of feedback mechanics. Challenges can be integrated into the game and the mechanics of competitions, especially if the game will be played by groups or through role-playing scenarios. Other game elements that would impact engagement in the game include avatars and levels.

### 7.2. Behaviour Change Theories

*'Self-Determination Theory'* was the most commonly used theory in 'Psychology' and 'Education' studies, and the second most common theory in 'Health' studies. This theory was adopted to develop games with aspects that are used to respond to the motivational factors related to autonomy, relatedness, and competence [170]. It has been adopted for health games to provide and improve wellbeing services [240]. This theory has also been adopted in many education games, for instance in games that help students practice music education [234], promotes students' motivation, engagement, and problem solving competencies [241], and its efficiency in motivating students through the social adaptive e-learning was proved [242].

*'Social Cognitive Theory'* was the most commonly adopted theory in 'Health' studies, and the third in 'Psychology' and 'Education' studies. Integrating this theory with health serious games has demonstrated game design that has an impact in directing and guiding healthy behaviours [243]. In psychology studies, it has been adopted to explore the video games' effect on players [244]. Education integrated with entertainment aspects was introduced by digital games through the adoption of social cognitive theory in the game design phases [245].

*'Theory of Planned Behaviour'* was the third most adopted theory in 'Health' studies, and the second in 'Psychology' and 'Education' studies. In health studies, it helps to inform game design that influences positive healthy behaviours, and has also been used to help predict healthy behaviours that can be achieved through games [36]. The theory allows the exploration of people's intentions to play online games [38]. It has contributed to education studies, as a game informed by this theory has been shown to stimulate children to use exergames for learning physical lessons [246].

*'Flow Theory'* was the third most frequently adopted theory in 'Health' and 'Psychology' studies. Flow theory can be used to design serious games able to help treat cognitive disorders in the elderly [115]. It has also led psychological games to investigate the effects of gamification [138].

*'Theory of Fun'* was the fourth most frequently mentioned and adopted theory in 'Education' studies; however, it did not appear in 'Health' or 'Psychology' studies. This theory has contributed to the design of games for education purposes [112].

*'Motivational Theory'* was ranked fourth in 'Health' studies, fifth in 'Psychology', and third in 'Education' studies. It was adopted to improve the travel behaviour change [108]. It was also part of designing gamified systems for health and wellbeing purposes [234]. Moreover, it was integrated into games that helped to practice music for students [234].

*'Self-Participation Theory'* was ranked fourth in 'Health' studies and third in 'Psychology' studies. It was used to explore players' attitudes towards using wheelchairs as part of health game applications [117]. Moreover, it was adopted to change behaviours by promoting pro-environmental behaviours [113].

## 8. Conclusions

This systematic review identified game features that are commonly or mostly used in studies associated with the different application domains that explore, redirect, and influence human behaviours. The applications covered by the review include serious games and gamified systems. The review included the examination of the use of game features and elements in 42 games, found in 15 fields. The results of the preliminary systematic review showed three main fields, namely 'Health', 'Psychology', and 'Education', that have reported more studies associated to the use serious games and/or gamified applications for behaviour change. A deeper review was carried out on the studies associated with these three dominant fields and the game elements associated with the game-based intervention used.

Our review and analysis revealed that each domain or context has a different priority of what game elements or features that were considered in the design, which are informed by the desired outcomes concerning the target users' attitude and/or behaviour. After conducting a focused systematic review on the three fields, we found that all the studies had similar presences but with differing priorities on the elements considered.

This paper aimed to provide directions to serious games developers and researchers who undertake building persuasive games. It helps to point out the most influential game elements and mechanics adopted to gain the highest impact expected. The aim extends to highlight the most frequently applied game elements in each field, though these are not necessarily to be adopted on their own, but rather integrated with other game mechanics and elements according to the context of the game and what is required. This study provides insights on the game elements, the theories, and their impact on various studies that will be useful for other game designers, developers, and researchers.

While this paper introduced a comprehensive systematic review, there are some limitations. One limitation that should be considered is measuring the impact of these studies on people and defining the game elements according to the level of influence. Additionally, this paper included all practical papers that employ games or gamified systems and excluded theoretical papers representing the state of art of the serious games, which may provide more insights of games, gamified systems, and their impact on behaviour change.

**Supplementary Materials:** The following supporting information can be downloaded at: https://www.mdpi.com/article/10.3390/info13030142/s1, Table S1: Full Classification of Game Elements; Table S2: Full Classification of Behaviour Change Theories.

**Author Contributions:** Conceptualization, R.H.; methodology, R.H. and S.A.; software, R.H.; validation, R.H., S.A.; formal analysis, R.H. and S.A.; investigation, R.H.; resources, R.H.; data curation, R.H. and S.A.; writing—original draft preparation, R.H.; writing—review and editing, S.A.; visualization, R.H.; supervision, S.A.; funding acquisition, R.H. All authors have read and agreed to the published version of the manuscript.

**Funding:** This research received no external funding.

**Institutional Review Board Statement:** Not applicable.

**Informed Consent Statement:** Not applicable.

**Data Availability Statement:** Not applicable.

**Acknowledgments:** I would like to thank Russell White at Solent University for the consistent support in this research. We also would like to thank University of Essex for funding this study.

**Conflicts of Interest:** The authors declare no conflict of interest.

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
