# Peer review of "Serious Gaming for Behaviour Change: A Systematic Review"

_information, doi:10.3390/info13030142_

Round 1

Reviewer 1 Report

The article is well written and shows the authors' excellent knowledge of the subject.
My only comment relates to Fig 3 and Fig 4, where the categories on the y-axis are not properly labelled.

Author Response

Response to Reviewer 2:

We would like to take the chance and thank you for your efforts to make the paper in the best shape. We really appreciated your review, and we took it significantly and this document shows the response of your beneficial comments.

Page

Line

Change requested

Response

1

10

“This is due to the intrinsic motivation that games afford”

This sentence makes no sense. Please rephrase with simpler words so everyone can understand

The sentence has been changed to:

This is due to the significance of the players’ intrinsic motivation that is naturally generated to play games”

1

12

Games don't intervene. Why don't you write "Many review papers measure the effectiveness of the use of gaming on changing behaviours"? It's simpler and guaranteed to be understood

The sentence has been changed to:

Many review papers measure the effectiveness of the use of gaming on changing behaviours”

1

16

This is also potentially confusing. Instead of using synonyms, I urge you to say what you mean in the simplest words possible. It will benefit the MS. I would write here something like "This paper identifies key theories of behaviour change, ...". "Shedding light" can mean many different things and makes the sentence unclear...

The sentence has been changed to:

“This paper identifies key theories of”

1

21

The introduction contains good information but does not flow. It introduces a barrage of definitions while providing little to no context between concepts and the main aim of the MS.

I added a new paragraph to make it an interesting story. From line 22 to line 40.

2

49

"Stimuli" is not a verb. Perhaps you mean "stimulate"?

The word replaced to be ‘Stimulate’

1

49

I would prefer "making decisions" over "taking decisions"

The word was replaced a “making”

1

50

domains

Fields

1

51

Fields

Removed

2

52

This paragraph should be at the top of the introduction as one of the first building blocks to tell the story of the introduction

The Gamification section moved up with Serious gaming as they presented different terms that has similar meanings.

It moved to page 2, line 50

2

61

I would really stay away from using fluffy language. I don't know how to interpret "the other side of the coin". If BCG and persuasive games are the same, please just say so. If persuasive games are a subset of BCG, then say so. This is a common issue with some parts of the MS, and you are doing yourself a disservice trying to pick pretty English that muddles the meaning of the text.

The sentence has been changed to:

“BCG has the same nature of what is called persuasive games”

2

63

“Naturally, video games used to 37

adopt persuasive strategies from game design;”

Another sentence that is unclear, and should be rewritten to make its point in simple words. I don't understand this

Naturally, video games are used to implement persuasive strategies by utilising the power of mechanics and elements of the game design

2

66

reward

The word replaced to be ‘rewards’

2

74

"enlisting" is not the same as "listing" or "summarizing" (which is how I think you meant to use it). Please be careful using pretty synonyms. Really. I strongly prefer simple English to bring across complicated content

The word replaced to be ‘listed’

2

73

"Dynamics" are not further discussed in the MS. If it is not relevant to the MS, please do not raise the subject!

The sentence has been changed to:

Werbach and Hunter [18] categorize game features into Mechanics, and Components.

2

78

serious games or gamified systems

AH! So they are synonyms?! I wish I had known that from the start (and the use of only one of the two terms would have been sufficient). The simplicity principle also applies to alternatives for technical terms: it's better to use one term consistently if you want your readers to follow along; using several terms that mean more or less the same thing just risks readers getting confused

‘or gamified systems’ removed

2

81

“and they are less diffuse compared

to entertainment games [21].”

IF this is important to the MS, please explain. If it is not important, please remove. The statement is not obvious

I amended this sentence and I had to mention the comparison between the two types of games as I am trying to resolve the confusion between them.

“Generally, features and mechanics in SGs have not yet been characterised and defined, due to their complex nature compared to entertainment games [20].”

2

90

Persuasive games use some tools such as reduction, tunnelling, conditioning, suggestion, and self-monitoring [17]. Educational games that intend to change learners’ behaviour often incorporate different game elements to increase motivation and engagement, such as challenges, points, and gifts [24, 25].

This has already been said

REMOVED

2

91

“Designing a persuasive game for particular purposes requires thorough understanding of 76

the relevant behaviour change theories that can feed into the design process [27].”

This is a very important statement that should move up in the introduction to frame the story

Moved up in the new introduction paragraph – Line: 30

3

113

“Many games, in many disciplines, have been designed purposely for behavioural change”

This should be one of the opening sentences of the introduction - it's one of the core reasons why you conducted this study

It was moved to page 1 line 22

3

115

In to be replaced to Across

Across

3

116

“generally; [15] focus on persuasive games.”

What is this?

You are right, thanks for spotting it! I removed it. It seems like part of the remained sentence is supposed to be removed entirely.

3

121

enlighten

provide

3

122

with essential information

With additional insights

3

132

I would remove the * wildcards from the key words

Removed

3

152

Are you categorizing games by applied discipline? The actual categorization is not clear to me

This sentence modified to clarify the reason for having games categories:

“This study categorised games which were designed to change people’s behaviours and attitudes as a separate category away from the mainstream of normal entertaining games.”

3

154

“As mentioned earlier, previous studies did not categorise games according to their specific roles or claims/evidence around behaviour change.

This sentence was deleted:

“As mentioned earlier, previous studies did not categorise games according to their specific roles or claims/evidence around behaviour change.”

3

161

Categorisation of outcomes

Categorisation of Objects

3

162

outcome

Objective

3

164

other relevant outcomes

other relevant objectives

4

169

/Delivery

Deleted

4

170

THis is an awkward sentence. How about writing "Our study considered games for any platform: desktop computers, video game consoles, mobile phones, tables, online, arcade consoles, and head-mounted devices."

This sentence replaced to be:
“Our study considered games for any platform – desktop computers, video consoles, mobile phones, tablets, online games, arcade and head-mounted display.”

4

175

demonstrated

Deleted

6

196

Figure 3 misses essential Y-axis labels

I fixed the Y Axis labels

7

199

Figure 4 misses essential Y-axis labels

I fixed the Y Axis labels

8

210

Studies adopting the theory
Table 2, rename column 2 to "Key studies" too

Replaced by “Key Studies”

8

211

Section 5 reads well, nice

Thanks ?

10

295

No serious comments to Section 6. It reads well; well done

Thanks ?

10

This is an example of using different terms for the same thing. The terms "Field", "Discipline" and "Domain" are used throughout the MS to indicate the same thing. As these terms can potentially have different meanings, their mixed use can lead to confusion. I urge you to pick one and use it consistently. Please.

I change all ‘Disciplines’ to ‘fields’

10

303

Perhaps the choice of domains / fields / disciplines should be mentioned in the Methods section to set readers' expectations

I change all ‘Disciplines’ to ‘fields’

15

417

‘Domains’ - highlighted

I change all ‘Disciplines’ to ‘fields’

15

419

why 5 game features and 6 behaviour change theories? What motivated the 5 / 6 difference?

I added this part to clarify it:
“the top five game features and the top six behaviour change theories as a sample. The reason is to spot the light on the most influential game elements across most frequent three fields to explore the correlations that might occur between these fields and game features from a side and between domains and behaviour change theories from the other side.”

16

445

Yes, THIS are proper outcomes

Thanks ?

17

479

“Then, based on what is needed, the game design can be built.”

I would remove this. Games can only be built on clear specifications; that is obvious. Explicitly mentioning it here and not elsewhere may confuse readers

Sentence removed

19

598

By now I should understand the difference, but I still am not sure why there is a distinction made between SG and gamified systems. Does it matter? Introduction, please help me out ;-)

The introduction covered the difference between the mentioned terms

19

609

I would remove this; you are re-iterating what was already said. Instead, I would use this section to talk to SG developers and convince them why your study matters.

A new paragraph was added to show the significance of this paper to the SG game developers and researchers.

Reviewer 2 Report

Brief summary

This MS provides a review of recent serious games, with particular focus on deployed game design mechanisms and behavioural theory underlying the games, to influence change. The concept of the paper is interesting and a useful addition to the field, as it provides valuable insights in priorities and concepts underlying published serious games.

General comments

The article is interesting and useful, and it provides a solid contribution to the field. However, it suffers from a number of general issues througout that should be addressed before the MS can be published

Article

The biggest weakness of the article is the introduction, which is hard to follow due to lack of storyline. THe introduction now reads like a glossary of terms with little context between concepts, and does not provide a clera motivation for the work (unlike the abstract). This is, however, now very difficult to fix. The article gets easier to read once section 2 comes around, after which the theoretical basis of the paper becomes clearer and sound. 

English

I am not a native English speaker, and I sympathise with the authors for my inescapable comment "please have your document checked for English". Aside from small spelling errors such as singular / pleural confusion, there are overarching weaknesses that must be addressed.

  1. The authors seem to use synonyms for the same terms, which creates confusion. For instance, what is the difference beween "domain", "field" or "discipline"? Is there a practical difference between "gamified systems" and "Serious Games, or between "BCG and persuasive games"? This is not clearly and explicitly resolved in the MS, and leaves me a bit lost. I suggest to pick one term and use it consistently, and if there are practical difference, please make these EXPLICITLY clear
  2. The point above is made worse by the use of woolly, prettified English that sometimes makes sentences unclear (or even flips their meaning). I am a big fan of using simple language to convey complex matters. A scientific MS should be written to be understood by being as clear as possible. It does not have to be a piece of beautiful literature. Attached PDF highlights a number of issues

Structure

Technical papers follow a strict format that this MS deviates from. Especially the introduction of theory in section 5 would be a no-go for me, but given the need to explain specific concepts in the context of the review process, the chosen structure is probably warranted. I have asked the Editor for assistance here.

Scientific content

I cannot judge if the work and cited literature is comprehensive as I am not an expert in many of the applied fields, but the description of the review process alludes to an unbiased and thorough methodological approach. I am impressed with that. Also, I am not the reviewer to detect plagiarism as I am not familiar with most works. I hope that other reviewers offer more sensible input here.

In the attached PDF I highlight more generic MS issues that will hopefully make the MS better. It's good work, but it needs a bit of plastic surgery before it can roll out to the general public.

Author Response

Response to Reviewer 2:

We would like to make the change and thank you for your efforts to make the paper in the best shape. We really appreciated your review, and we took it significantly and this document shows the response of your beneficial comments.

Page

Line

Change requested

Response

1

10

“This is due to the intrinsic motivation that games afford”

This sentence makes no sense. Please rephrase with simpler words so everyone can understand

The sentence has been changed to:

This is due to the significance of the players’ intrinsic motivation that is naturally generated to play games”

1

12

Games don't intervene. Why don't you write "Many review papers measure the effectiveness of the use of gaming on changing behaviours"? It's simpler and guaranteed to be understood

The sentence has been changed to:

Many review papers measure the effectiveness of the use of gaming on changing behaviours”

1

16

This is also potentially confusing. Instead of using synonyms, I urge you to say what you mean in the simplest words possible. It will benefit the MS. I would write here something like "This paper identifies key theories of behaviour change, ...". "Shedding light" can mean many different things and makes the sentence unclear...

The sentence has been changed to:

“This paper identifies key theories of”

1

21

The introduction contains good information but does not flow. It introduces a barrage of definitions while providing little to no context between concepts and the main aim of the MS.

I added a new paragraph to make it an interesting story. From line 22 to line 40.

2

49

"Stimuli" is not a verb. Perhaps you mean "stimulate"?

The word replaced to be ‘Stimulate’

1

49

I would prefer "making decisions" over "taking decisions"

The word was replaced to be “making”

1

50

domains

Fields

1

51

Fields

Removed

2

52

This paragraph should be at the top of the introduction as one of the first building blocks to tell the story of the introduction

The Gamification section moved up with Serious gaming as they presented different terms that has similar meanings.

It moved to page 2, line 50

2

61

I would really stay away from using fluffy language. I don't know how to interpret "the other side of the coin". If BCG and persuasive games are the same, please just say so. If persuasive games are a subset of BCG, then say so. This is a common issue with some parts of the MS, and you are doing yourself a disservice trying to pick pretty English that muddles the meaning of the text.

The sentence has been changed to:

“BCG has the same nature of what is called persuasive games”

2

63

“Naturally, video games used to 37

adopt persuasive strategies from game design;”

Another sentence that is unclear, and should be rewritten to make its point in simple words. I don't understand this

Naturally, video games are used to implement persuasive strategies by utilising the power of mechanics and elements of the game design

2

66

reward

The word replaced to be ‘rewards’

2

74

"enlisting" is not the same as "listing" or "summarizing" (which is how I think you meant to use it). Please be careful using pretty synonyms. Really. I strongly prefer simple English to bring across complicated content

The word replaced to be ‘listed’

2

73

"Dynamics" are not further discussed in the MS. If it is not relevant to the MS, please do not raise the subject!

The sentence has been changed to:

Werbach and Hunter [18] categorize game features into Mechanics, and Components.

2

78

serious games or gamified systems

AH! So they are synonyms?! I wish I had known that from the start (and the use of only one of the two terms would have been sufficient). The simplicity principle also applies to alternatives for technical terms: it's better to use one term consistently if you want your readers to follow along; using several terms that mean more or less the same thing just risks readers getting confused

‘or gamified systems’ removed

2

81

“and they are less diffuse compared

to entertainment games [21].”

IF this is important to the MS, please explain. If it is not important, please remove. The statement is not obvious

I amended this sentence and I had to mention the comparison between the two types of games as I am trying to resolve the confusion between them.

“Generally, features and mechanics in SGs have not yet been characterised and defined, due to their complex nature compared to entertainment games [20].”

2

90

Persuasive games use some tools such as reduction, tunnelling, conditioning, suggestion, and self-monitoring [17]. Educational games that intend to change learners’ behaviour often incorporate different game elements to increase motivation and engagement, such as challenges, points, and gifts [24, 25].

This has already been said

REMOVED

2

91

“Designing a persuasive game for particular purposes requires thorough understanding of 76

the relevant behaviour change theories that can feed into the design process [27].”

This is a very important statement that should move up in the introduction to frame the story

Moved up in the new introduction paragraph – Line: 30

3

113

“Many games, in many disciplines, have been designed purposely for behavioural change”

This should be one of the opening sentences of the introduction - it's one of the core reasons why you conducted this study

It was moved to page 1 line 22

3

115

In to be replaced to Across

Across

3

116

“generally; [15] focus on persuasive games.”

What is this?

You are right, thanks for spotting it! I removed it. It seems like part of the remained sentence that is supposed to be removed entirely.

3

121

enlighten

provide

3

122

with essential information

With additional insights

3

132

I would remove the * wildcards from the key words

Removed

3

152

Are you categorizing games by applied discipline? The actual categorization is not clear to me

This sentence modified to clarify the reason for having games categories:

“This study categorised games which were designed to change people’s behaviours and attitudes as a separate category away from the mainstream of normal entertaining games.”

3

154

“As mentioned earlier, previous studies did not categorise games according to their specific roles or claims/evidence around behaviour change.

This sentence was deleted:

“As mentioned earlier, previous studies did not categorise games according to their specific roles or claims/evidence around behaviour change.”

3

161

Categorisation of outcomes

Categorisation of Objects

3

162

outcome

Objective

3

164

other relevant outcomes

other relevant objectives

4

169

/Delivery

Deleted

4

170

THis is an awkward sentence. How about writing "Our study considered games for any platform: desktop computers, video game consoles, mobile phones, tables, online, arcade consoles, and head-mounted devices."

This sentence replaced to be:
“Our study considered games for any platform – desktop computers, video consoles, mobile phones, tablets, online games, arcade and head-mounted display.”

4

175

demonstrated

Deleted

6

196

Figure 3 misses essential Y-axis labels

I fixed the Y Axis labels

7

199

Figure 4 misses essential Y-axis labels

I fixed the Y Axis labels

8

210

Studies adopting the theory
Table 2, rename column 2 to "Key studies" too

Replaced by “Key Studies”

8

211

Section 5 reads well, nice

Thanks ?

10

295

No serious comments to Section 6. It reads well; well done

Thanks ?

10

This is an example of using different terms for the same thing. The terms "Field", "Discipline" and "Domain" are used throughout the MS to indicate the same thing. As these terms can potentially have different meanings, their mixed use can lead to confusion. I urge you to pick one and use it consistently. Please.

I change all ‘Disciplines’ to ‘fields’

10

303

Perhaps the choice of domains / fields / disciplines should be mentioned in the Methods section to set readers' expectations

I change all ‘Disciplines’ to ‘fields’

15

417

‘Domains’ - highlighted

I change all ‘Disciplines’ to ‘fields’

15

419

why 5 game features and 6 behaviour change theories? What motivated the 5 / 6 difference?

I added this part to clarify it:
“the top five game features and the top six behaviour change theories as a sample. The reason is to spot the light on the most influential game elements across most frequent three fields to explore the correlations that might occur between these fields and game features from a side and between domains and behaviour change theories from the other side.”

16

445

Yes, THIS are proper outcomes

Thanks ?

17

479

“Then, based on what is needed, the game design can be built.”

I would remove this. Games can only be built on clear specifications; that is obvious. Explicitly mentioning it here and not elsewhere may confuse readers

Sentence removed

19

598

By now I should understand the difference, but I still am not sure why there is a distinction made between SG and gamified systems. Does it matter? Introduction, please help me out ;-)

The introduction covered the difference between the mentioned terms

19

609

I would remove this; you are re-iterating what was already said. Instead, I would use this section to talk to SG developers and convince them why your study matters.

A new paragraph was added to show the significance of this paper to the SG game developers and researchers.

Reviewer 3 Report

In their submission with the title “Serious Gaming for Behaviour Change: A Systematic Review”, the authors provide a broad overview of works published in this area and also analyse the works identified for commonly used game elements and for commonly employed behaviour change theories (BCT).

The outcomes indicate an general increasing trend in numbers if publications in this area with differences in the frequencies observed across application use cases.

The work addresses a topic that is of considerable potential relevance to a growing community of researchers and practitioners.

The general motivational framing, setup, analysis, etc. are described in a clear an easy-to-follow manner. At times – yet throughout the work – conclusions and generalisations can be relatively broad, appearing not always fully supported by clear evidence reported (e.g. “Many review papers measure the effectiveness of game interventions on changing behaviours; however, these studies neglect the game features involved in the game design process, which have an impact of stimulating behaviour change.”).

Structure and language generally clear and easy to follow, but frequent grammar and phrasing errors (with some spelling mistakes) are still present.

The introduction on gamification is not clearly framed regarding the relationship the authors see between this term and the terms behaviour change games and persuasive games. Likewise, links between a range of theories of behavior change are loosely indicated in section 1, but not made explicit.

An effectiveness analysis (e.g. considering reported outcomes regarding BC depending on specific game elements of BCTs being employed), which is indicated in the introduction, is not delivered in the analysis if this work.

While the search terms appear reasonable, the motivation for the search composition (in particular the AND part) is not clear and the repetition of some terms in the AND part is somewhat unusual (i.e. warrants explanation). It is not clear whether the authors performed any further scanning of related work by considering e.g. lists of references of primary hits and again: how the search query was constructed / motivated, as well as checked for validity. A clear line of argument around search terms appears especially relevant given another key limitation that warrants discussion: the topic area is arguably concerned with a considerably broad range of terms in a (relatively) young and hence arguably less-solidified research area.

Another important methodological clarification is missing wrgt. how the classifications were performed and verified (e.g. for game elements present, behavior change theories implemented, etc.). It is, however, commendable that a full classification table is provided as an attachment.

Some data presented in tables and figures makes either respective display somewhat obsolete (e.g. it is debatable whether Figure 4 really needed; and if it is found to be needed then this begs the question of the specific purpose it serves).

Although the work references a considerable number of BCTs that are indeed commonly applied, the key working principles, as well as the similarities, differences and common realisations of the BCTs in  serious games are only vaguely illustrated. In addition, it appears that “Motivational Theory” (cf. e.g. Table 2) is not truly a BCT, but rather a general reference term used in other works. The same likely applies to “theory of fun”. At the same time, Flow and the SDT are arguably a theories of motivation and not BC. Therefore, I think the work would benefit from an improved differentiating discussion of the various theories.

Concerning a structural question, the separation between health and “psychology domain” (possibly akin to “mental health”) is not clear (how what is meant is different and also how the authors arrived at this).

Overall, the work presents a clear research aim and an interesting descriptive discussion of the outcomes. However, in the current state it lacks precision on specific theories employed and a contextualization of the categorisations. Drawing the points discussed above together, I tend slightly towards recommending major revisions (as compared to the alternative of minor revision).

Specific comments

Labeling of categories in Figure 3 is incomplete.

Table 1: consider explicitly linking to attachments for complete listings of occurrences.

Table 2: mark “Studies adopting the theory” as e.g.?

Additional references to consider:

https://www.igi-global.com/article/a-critical-review-of-the-effectiveness-of-narrative-driven-digital-educational-games/213970

Author Response

Response to All Reviewers:

We would like to take the chance and thank you all for your efforts to make the paper in the best shape. We really appreciated your reviews, and we took in significantly and this document shows the response to your beneficial comments.

RVWR: Reviewers           P: Page                L: Line

RVWR

P

L

Change requested

Response

2

1

10

“This is due to the intrinsic motivation that games afford”

This sentence makes no sense. Please rephrase with simpler words so everyone can understand

The sentence has been changed to:

This is due to the significance of the players’ intrinsic motivation that is naturally generated to play games”

2

1

11

Games don't intervene. Why don't you write "Many review papers measure the effectiveness of the use of gaming on changing behaviours"? It's simpler and guaranteed to be understood

The sentence has been changed to:

Many review papers measure the effectiveness of the use of gaming on changing behaviours”

2

1

16

This is also potentially confusing. Instead of using synonyms, I urge you to say what you mean in the simplest words possible. It will benefit the MS. I would write here something like "This paper identifies key theories of behaviour change, ...". "Shedding light" can mean many different things and makes the sentence unclear...

The sentence has been changed to:

“This paper identifies key theories of”

2

1

21

The introduction contains good information but does not flow. It introduces a barrage of definitions while providing little to no context between concepts and the main aim of the MS.

I added a new paragraph to make it an interesting story. From line 22 to line 40.

3

2

21

The introduction on gamification is not clearly framed regarding the relationship the authors see between this term and the terms behaviour change games and persuasive games. Likewise, links between a range of theories of behavior change are loosely indicated in section 1, but not made explicit.

I added a new paragraph to make it an interesting story about the topic then I created a connection between the terms used and the behaviour changes games and I highlighted the relation between them. From line 22 to line 40. Then, I amended the title and content of section 1.1. Serious Games & Gamification to include all terms in it.

2

2

49

"Stimuli" is not a verb. Perhaps you mean "stimulate"?

The word replaced to be ‘Stimulate’

2

2

49

I would prefer "making decisions" over "taking decisions"

The word replaced to be “making”

2

2

50

domains

Fields

2

2

51

Fields

Removed

2

2

52

This paragraph should be at the top of the introduction as one of the first building blocks to tell the story of the introduction

Gamification section moved up with Serious gaming as they presenting different terms has similar meanings.

It moved to page 2, line 52

2

2

61

I would really stay away from using fluffy language. I don't know how to interpret "the other side of the coin". If BCG and persuasive games are the same, please just say so. If persuasive games are a subset of BCG, then say so. This is a common issue with some parts of the MS, and you are doing yourself a disservice trying to pick pretty English that muddles the meaning of the text.

The sentence has been changed to:

“BCG has the same nature of what are called persuasive games”

2

2

63

“Naturally, video games used to 37

adopt persuasive strategies from game design;”

Another sentence that is unclear, and should be rewritten to make its point in simple words. I don't understand this

Naturally, video games used to implement persuasive strategies by utilising the power of mechanics and elements of the game design

2

2

66

reward

The word replaced to be ‘rewards’

2

2

74

"enlisting" is not the same as "listing" or "summarizing" (which is how I think you meant to use it). Please be careful using pretty synonyms. Really. I strongly prefer simple English to bring across complicated content

The word replaced to be ‘listed’

2

2

73

"Dynamics" are not further discussed in the MS. If it is not relevant to the MS, please do not raise the subject!

The sentence has been changed to:

Werbach and Hunter [18] categorize game features into Mechanics, and Components.

2

2

78

serious games or gamified systems

AH! So they are synonyms?! I wish I had known that from the start (and the use of only one of the two terms would have been sufficient). The simplicity principle also applies to alternatives for technical terms: it's better to use one term consistently if you want your readers to follow along; using several terms that mean more or less the same thing just risks readers getting confused

‘or gamified systems’ removed

2

2

81

“and they are less diffuse compared

to entertainment games [21].”

IF this is important to the MS, please explain. If it is not important, please remove. The statement is not obvious

I amended this sentence and I had to mention the comparison between the two types of games as I am trying to resolve the confusion between them.

“Generally, features and mechanics in SGs have not yet been characterised and defined, due to their complex nature compared to entertainment games [20].”

2

2

90

Persuasive games use some tools such as reduction, tunnelling, conditioning, suggestion, and self-monitoring [17]. Educational games that intend to change learners’ behaviour often incorporate different game elements to increase motivation and engagement, such as challenges, points, and gifts [24, 25].

This has already been said

REMOVED

2

2

91

“Designing a persuasive game for particular purposes requires thorough understanding of 76

the relevant behaviour change theories that can feed into the design process [27].”

This is a very important statement that should move up in the introduction to frame the story

Moved up in the new introduction paragraph – Line: 30

3

3

113

The general motivational framing, setup, analysis, etc. are described in a clear an easy-to-follow manner. At times – yet throughout the work – conclusions and generalisations can be relatively broad, appearing not always fully supported by clear evidence reported (e.g. “Many review papers measure the effectiveness of game interventions on changing behaviours; however, these studies neglect the game features involved in the game design process, which have an impact of stimulating behaviour change.”).

This section has been added to the problem statement section:

“Many review papers measure the effectiveness of game interventions on changing behaviours; however, these studies neglect the game features involved in the game design process, which have an impact of stimulating behaviour change.”

2

3

113

“Many games, in many disciplines, have been designed purposely for behavioural change”

This should be one of the opening sentences of the introduction - it's one of the core reasons why you conducted this study

It was moved to page 1 line 22

2

3

118

In to be replaced to Across

Across

2

3

118

“generally; [15] focus on persuasive games.”

What is this?

You are right, thanks spotting it! I removed it. It seems like part of remained sentence that it supposed to be removed entirely.

2

3

124

enlighten

provide

2

3

125

with essential information

With additional insights

2

3

137

I would remove the * wildcards from the key words

Removed

3

3

136

While the search terms appear reasonable, the motivation for the search composition (in particular the AND part) is not clear and the repetition of some terms in the AND part is somewhat unusual (i.e. warrants explanation).

The only (AND) exists in line 136 between Computer Games AND Change attitude to find studies that used SG to change behaviours and the justification of that already explained in the previous section under 1.5 Problem statement in “This paper aims to highlight the most effective game elements that have a significant effect on players’ behaviour.”

2

4

156

Are you categorizing games by applied discipline? The actual categorization is not clear to me

This sentence modified to clarify the reason of having games categories:

“This study categorised games which were designed to change people’s behaviours and attitudes as a separate category away from the mainstream of normal entertaining games.”

2

4

158

“As mentioned earlier, previous studies did not categorise games according to their specific roles or claims/evidence around behaviour change.

This sentence deleted:

“As mentioned earlier, previous studies did not categorise games according to their specific roles or claims/evidence around behaviour change.”

2

4

164

Categorisation of outcomes

Categorisation of Objects

2

4

167

outcome

Objective

2

4

167

other relevant outcomes

other relevant objectives

2

4

169

/Delivery

Deleted

2

5

173

THis is an awkward sentence. How about writing "Our study considered games for any platform: desktop computers, video game consoles, mobile phones, tables, online, arcade consoles, and head-mounted devices."

This sentence replaced to be:
“Our study considered games for any platform – desktop computers, video consoles, mobile phones, tablets, online games, arcade and head-mounted display.”

2

5

178

demonstrated

Deleted

3

5

194

An effectiveness analysis (e.g. considering reported outcomes regarding BC depending on specific game elements of BCTs being employed), which is indicated in the introduction, is not delivered in the analysis if this work.

The analysis that connects between game elements and BCTs has been manifested in the following paragraph: “Analysis of this mapping could allow for investigation into the rationale behind the adoption of these elements in the fields that have shown the highest adoption of BCG studies.”

It also extends in the discussion section to demonstrate the correlations between the frequencies between the game elements and the most adopted BCTs in Serious games to direct researchers and developers with the most influential tools and methods to build persuasive and impactful games.

1,2&3

6

200

Figure 3 misses essential Y-axis labels

Labeling of categories in Figure 3 is incomplete

I fixed the Y Axis labels

3

6

201

Table 1: consider explicitly linking to attachments for complete listings of occurrences.

[There is a supplementary attachment to the article that demonstrates the classification and definitions of game elements]

1&2

7

203

Figure 4 misses essential Y-axis labels

I fixed the Y Axis labels

2

8

214

Studies adopting the theory
Table 2, rename column 2 to "Key studies" too

Here I have 2 different comments from 2 different reviewer with all respect, I think the best answer is to name it as “Examples of studies adopt the theory”

3

8

214

Table 2: mark “Studies adopting the theory” as e.g.?

2

8

215

Section 5 reads well, nice

Thanks ?

2

10

300

No serious comments to Section 6. It reads well; well done

Thanks ?

2

10

All

This is an example of using different terms for the same thing. The terms "Field", "Discipline" and "Domain" are used throughout the MS to indicate the same thing. As these terms can potentially have different meanings, their mixed use can lead to confusion. I urge you to pick one and use it consistently. Please.

I change all ‘Disciplines’ to ‘fields’

3

10

315

Concerning a structural question, the separation between health and “psychology domain” (possibly akin to “mental health”) is not clear (how what is meant is different and also how the authors arrived at this).

Psychological Studies that came into the scope of papers of the study did not encounter any subject to metal health. It is more like changing attitudes of game players, exergames, sports or preventing behaviours e.g. smoking  

2

10

303

Perhaps the choice of domains / fields / disciplines should be mentioned in the Methods section to set readers' expectations

I change all ‘Disciplines’ to ‘fields’

3

13

379

Additional references to consider:

https://www.igi-global.com/article/a-critical-review-of-the-effectiveness-of-narrative-driven-digital-educational-games/213970

A recent significant critical review study showed the effectiveness of using the educational games compared to the traditional methods in addition to promoting attitude change [197].

2

15

417

‘Domains’ - highlighted

I change all ‘Disciplines’ to ‘fields’

2

15

426

why 5 game features and 6 behaviour change theories? What motivated the 5 / 6 difference?

I added this part to clarify it:
“the top five game features and the top six behaviour change theories as a sample. The rea-son is spot the light on the most influential game elements across most frequent three fields to explore the correlations that might occur between these fields and game features from a side and between domains and behaviour change theories from the other side.”

2

16

450

Yes, THIS are proper outcomes

Thanks ?

2

17

479

“Then, based on what is needed, the game design can be built.”

I would remove this. Games can only be built on clear specifications; that is obvious. Explicitly mentioning it here and not elsewhere may confuse readers

Sentence removed

2

19

606

By now I should understand the difference, but I still am not sure why there is a distinction made between SG and gamified systems. Does it matter? Introduction, please help me out ;-)

The introduction covered the difference between the mentioned terms

2

19

618

I would remove this; you are re-iterating what was already said. Instead, I would use this section to talk to SG developers and convince them why your study matters.

A new paragraph added to show the significance of this paper to the SG game developers and researchers.

3

All

All

Another important methodological clarification is missing wrgt. how the classifications were performed and verified (e.g. for game elements present, behavior change theories implemented, etc.). It is, however, commendable that a full classification table is provided as an attachment.

We made an attachment that defines and classify the Game elements and the Behaviour Change theories and their definitions to clear the confusion for the reader. It is submitted as a second attachment supplementary file,

3

All

All

Structure and language generally clear and easy to follow, but frequent grammar and phrasing errors (with some spelling mistakes) are still present.

The manuscript has been proofread and checked grammatically

Round 2

Reviewer 2 Report

Many thanks for the effort you put in. I hope you found my suggestions helpful, and I do think that the MS reads much better. Well done.

I have a few remaining suggestions all related to formatting and word use, but I have absolutely no problems with the scientific content of the work. I like the paper and would love to see it published. Well done.

My comments:

1. Introduction

22-23: I would repharase the first sentence to start stronger, such as: "In the last four decades, games designed purposely for behavioural change [1,2] are increasingly deployed in a wide range of professional settings.". 

23: Games are pleural. "its" -> "their"

24: How about "Researches exploited the appeal of playing games"?

27: I would remove "and often reported". The sentence flows better without these three words.

28: Remove "Furthermore, ". As a suggestion: "New terms have emerged to defined these types of games such as: serious games, gamified approaches, persuasive games, etc."

33: I would not mix tenses in the introduction. I would replace "were" with "are"

35: I think the words "work" or "manuscript" are more approproate than "paper"

36: "..., the behavioural change theories..."

36-37: Split the sentence. ".. to structure these games. This will provide ..."

38: Replace "The paper" with "The review"

42: Note ".,"

43: Add "."

48-52: please be consistent with the use of singular and pleural, even in definitions. Serious Games are interactive computer applications, that have goals, are fun to play, and supply skills... They are also "Games designed for ...". Perhaps this is nitpicking, but I think it's much easier on the reader.

Sentence 56-58 does not really flow. Perhaps change "refers" with "to refer", but I am not sure what you want to say here. Please rethink and rephrase.

Paragraph 69 - 82 is fantastic, well done.

I would conclude paragraph 1.2 with a sentence that explicitly states that your review focusses on BCG. This provides clarity and avoids potential confusion onwards.

117: You alread clarified that BCG are persuasive games. Don't make the reader doubt; start this sentence as "Various BCG from different ...". Simpler, valid, and less chance for confusion. Would you agree? The rest of section 1.4 is beautiful and to the point.

Section 1.5 is great too. Flows like a charm; I am impressed.

2. Method(s)

156: Method or Methods? Not sure, depends on the journal. Pleural is more common (as you rarely use only one method). 

For the rest section 2 is great. No comments. The figure is very helpful.

3. Data analysis

183-185: These sentences does not flow well. How about: "In line with other studies [29, 51, 52], we followed the Werbach and Hunter classification of game elements by mechaniscs and components [28]." Or something like that.

188: I would replace "So" with "Thus"; it's less colloquial. But more boring.

196-197: How about a small change: "The main objective ... whether published games aimed to change behaviour, and through which design choices." ?

4. Results

219: I'd change the first sentence to something like "Our review quantified the number of times that specific game elements were explicitly mentioned. As depicted in Table 1, ..."

221-222: Perhaps rephrase the supplementary attachment sentence to "For more details, please Table SXXX in the supplementary material."

223-226: I think this sentence is meant to say that some features were mentioned more than others. I think that sentence 226-228 si actually better placed a neat discussion point for future work, and could be moved to its own short paragraph in the Discussion? What do you think?

Thanks for fixing the table axis labels throughout the MS!

5. General findings

A strong section! Some suggestions:

258: "The reward system is thus a motivational ..."

282  How about "It can also be used as a strategy to increase game play motivation in pursuit of short-term rewards [170]." ?

299-300: Strange sentence. Perhaps replace "investigating the sense" with "stimulating"? Violition is an impressive term.

315: "embraced" is quite dramatic. How about replace with simple "used"?

318: another embrace here. Consider replacing with something more factual, the word is out of place.

6. Findings..

337: Suggest to remove "Therefore, " as it is not really needed and was used in the sentence prior.

342-343:  How about "The next sections outline the motiviations behind using BCG in the fields of health, psychology and education."?

407: ".. the rationale that young people are very familiar with games and gameplay, which .."

7. Discussion

Table 3: the feature count is quite meaningless as there is no real way to compare if features are used more or less between the different domains. You can solve this by either adding a row "Total studies", and list the total number of papers that address each specific domain, or by turning the feature count into a percentage of studies for each domain (e.g., Health / Challenges (X%), etc). This provides readers with the means to understand which field prioritizes specific game features, is strengthens the comparison section 7.1.

Section 7 contains some formatting inconsistencies. Section 7.1 introduce topics in single brackets (e.g., 'Challenges', 'Levels') while section 7.2 use italics (e.g., Self-Determination Theory, Social Cognitive Theory). Nitpicking, but easy to standardize

457-459: the sentence "Table 3 shows ... as a sample" is not clear and perhaps is not necessary. Suggestion: remove the sentence, and instead, start sentence 462-465 with "Table 3 reflects what is presented in ...".

8. Conclusions

As mentioned above, perhaps also put a note in for follow up work.

Author Response

Response to Reviewer 2:

We would like to take the chance and thank you for your efforts to make the paper in the best shape. We really appreciated your reviews, and we took in significantly and this document shows the response to your beneficial comments

Reviewer 3 Report

I would like to thank the authors for considering my prior remarks and for investing notable effort into improving the paper. I appreciate some of the general clarifications, minor language improvements, and in particular the extended attachment detailing and providing references for the relevant BCTs.

However, I still find many of my prior points remain to be addressed, at least to some - and in some cases to considerable - degree.

The text is still riddled with frequent grammatical and phrasing mistakes (while also containing some - but relatively few - spelling mistakes). This requires revision. In particular the use of some of the relevant terminology indicates potential misunderstandings that require clarification. E.g. the use of the term "psychology". I would like to ask the authors to clarify whether they mean "psychology" (as in either a research subject, or an aspect of human existence), or whether they mean "mental health" or "mental wellbeing", as aspects of human health and mental state that are frequently addressed by behaviour change interventions.

Below, I will address some of the author responses to my prior remarks in turn, quoting from the table as:
RVWR;P;L;Change requested;Response
-->
Comment / further remarks based on response...

3; 2; 21; The introduction on gamification is not clearly framed regarding the relationship the authors see between this term and the terms behaviour change games and persuasive games. Likewise, links between a range of theories of behavior change are loosely indicated in section 1, but not made explicit.; I added a new paragraph to make it an interesting story about the topic then I created a connection between the terms used and the behaviour changes games and I highlighted the relation between them. From line 22 to line 40. Then, I amended the title and content of section 1.1. Serious Games & Gamification to include all terms in it.
--> 
I recognise an improvement was made, but would still suggest the authors to start out with a clear definition of the different terms, as they are not equivalent or commonly used interchangeably.

3;3;113;The general motivational framing, setup, analysis, etc. are described in a clear an easy-to-follow manner. At times – yet throughout the work – conclusions and generalisations can be relatively broad, appearing not always fully supported by clear evidence reported (e.g. “Many review papers measure the effectiveness of game interventions on changing behaviours; however, these studies neglect the game features involved in the game design process, which have an impact of stimulating behaviour change.”).;This section has been added to the problem statement section: “Many review papers measure the effectiveness of game interventions on changing behaviours; however, these studies neglect the game features involved in the game design process, which have an impact of stimulating behaviour change.”
-->
Here I would have hoped to see examples of work referenced that can be seen to neglect the consideration of game features involved in the game design process to support the claim made. Similar generalisations are still frequent throughout the work.

3;3;136;While the search terms appear reasonable, the motivation for the search composition (in particular the AND part) is not clear and the repetition of some terms in the AND part is somewhat unusual (i.e. warrants explanation).;The only (AND) exists in line 136 between Computer Games AND Change attitude to find studies that used SG to change behaviours and the justification of that already explained in the previous section under 1.5 Problem statement in “This paper aims to highlight the most effective game elements that have a significant effect on players’ behaviour.”
-->
I thank the authors for the clarification. To me this still begs the question how e.g. the consideration of domain-specific impact goals for behavior change (e.g. improving adherence in healt-related playful or gameful interventions) were excluded / considered not to be relevant. I.e. what I was asking for was clarification on the process of how the search was composed (and possibly iterated / refined / reviewed).

3;5;194;An effectiveness analysis (e.g. considering reported outcomes regarding BC depending on specific game elements of BCTs being employed), which is indicated in the introduction, is not delivered in the analysis if this work.;The analysis that connects between game elements and BCTs has been manifested in the following paragraph: “Analysis of this mapping could allow for investigation into the rationale behind the adoption of these elements in the fields that have shown the highest adoption of BCG studies.”;It also extends in the discussion section to demonstrate the correlations between the frequencies between the game elements and the most adopted BCTs in Serious games to direct researchers and developers with the most influential tools and methods to build persuasive and impactful games.
-->
In the current state I still do not see clear indications of an effectiveness analysis and would suggest the authors to revise the indicated aims/outcomes accordingly. Arguably, an effectiveness analysis would require statistical analysis of common outcome variables across multiple identified publications, or at least a statistical treatment of the "correlations" indicated by the authors, which are - in the current version - only implicitly visible by comparing across the application areas highlighted by the authors. Even if the authors decide to have a qualitative focus, one would expect some structured empirical comparison between the different features and their hypothesised impacts as encountered or even expected by the authors of the works identified and considered in this review.

3;All;All;Another important methodological clarification is missing wrgt. how the classifications were performed and verified (e.g. for game elements present, behavior change theories implemented, etc.). It is, however, commendable that a full classification table is provided as an attachment.;We made an attachment that defines and classify the Game elements and the Behaviour Change theories and their definitions to clear the confusion for the reader. It is submitted as a second attachment supplementary file,
-->
As I noted above: the attachment is appreciated and marks a clear improvement to the work. However, it is still not clear to me what process the authors used to perform the classification (i.e. how was it determined which classes to use; who was involved in the process; was it verified somehow; etc).

Author Response

Response to Reviewer 3:

We would like to take the chance and thank you all for your efforts to make the paper in the best shape. We really appreciated your reviews, and we took in significantly and this document shows the response to your beneficial comments.

Round 3

Reviewer 3 Report

I thank the authors for their detailed response and find the manuscript to be notably improved as detailed in the prior process.

Author Response

Thanks again for your efforts, it is now in better shape.